# Systematic review to evaluate accuracy studies of the diagnostic criteria for periodontitis in pregnant women

**Sarah dos Santos Conceição**[1], **Josicélia Estrela Tuy Batista**[2], **Simone Seixas da Cruz**[2,3], **Isaac Suzart Gomes-Filho**[2]*, **Alexandre Marcelo Hintz**[2], **Julita Maria Freitas Coelho**[2], **Johelle de Santana Passos-Soares**[2,4], **Peter Michael Loomer**[5], **Amanda Oliveira Lyrio**[1], **Elivan Silva Souza**[1], **Ana Cláudia Morais Godoy Figueiredo**[6], **Mauricio Gomes Pereira**[1]

1 Faculty of Health Sciences, University of Brasilia, Brasília, Distrito Federal, Brazil, 2 Department of Health, Feira de Santana State University, Feira de Santana, Bahia, Brazil, 3 Health Sciences Center, Federal University of Recôncavo of Bahia, Cruz das Almas, Bahia, Brazil, 4 Department of Preventive Dentistry, Federal University Bahia, Salvador, Bahia, Brazil, 5 School of Dentistry, University of Texas Health Science Center at San Antonio, San Antonio, Texas, United States of America, 6 Epidemiology Surveillance, Federal District Health State Secretariat, Brasília, Bahia, Distrito Federal, Brazil

* isuzart@gmail.com

**Data Availability Statement:** All relevant data are within the manuscript and its Supporting Information files.

## Abstract

The diversity of criteria used in the diagnosis of periodontitis in pregnant women makes it difficult to define and compare the disease. Using a systematic review, this study evaluated the accuracy of criteria for diagnosing periodontitis in pregnant women. Searches were carried out in the databases: Medline/PubMed, Embase, Scopus, Web of Science, SciELO, Lilacs, ProQuest, and CINAHL. Validation studies of the criteria used for diagnosing periodontitis met the inclusion criteria. The study steps were performed by three independent reviewers. A qualitative synthesis of the included articles was carried out and the quality methodological analysis using the QUADAS-2 instrument. The assessment of the quality of the evidence was obtained through the GRADE tool. A total of 592 records were identified, of which only 6 made up this systematic review. The included studies analyzed different criteria for diagnosing periodontitis, evaluating 3,005 pregnant women. The criteria selected as a test presented results equivalent to the performance of those recognized as the gold standard. The self-reported criteria were of lower diagnostic accuracy. A major limitation of this review was the small number of primary studies that evaluated clinical diagnosis of periodontitis in pregnant women, which was highly heterogeneous, making it impossible to carry out accuracy meta-analysis and additional analyzes. There is a lack of consensus on the criteria for the diagnosis of periodontitis, with great variability in the accuracy and prevalence of the disease in pregnant women.

## Introduction

Periodontitis is the second most prevalent oral disease, affecting more than 50% of the adult population in the world [1, 2]. From the 1990s, population data have shown that the

**Funding:** The Coordination for the Improvement of Higher Education Personnel, number 001, with the granting of a doctoral scholarship to Dr. Sarah dos Santos Conceição (Coordination for the Improvement of Higher Education Personnel – CAPES), Federal District Research Support Foundation (Fundação de Apoio a Pesquisa do Distrito Federal), Distrito Federal, Brasilia, Brazil, and University of Brasilia, Distrito Federal, Brasilia, Brazil provided financial support for the research. The funder did not play any role in the study design, data collection and analysis, decision to publish, or preparation of the manuscript.

**Competing interests:** The authors have declared that no competing interests exist.

magnitude of the disease frequency differs between countries [3, 4]. It is known that the diversity of criteria used for the diagnosis of periodontitis makes it difficult, in both defining disease and comparing between studies [5].

Periodontitis is a chronic, immunoinflammatory, infection, multifactorial that shares common risk factors with several systemic conditions [6]. The presence of periodontal pathogens and their metabolic byproducts can modulate the host immune response beyond the oral cavity and lead to systemic complications, including adverse pregnancy outcomes, such as prematurity, low birth weight, and preeclampsia, a common health problem consisting of several health effects involving pregnancy and the newborn infant [7]. Due to hormonal changes, pregnant women are more susceptible to infections, but pregnancy does not cause periodontitis. However, pregnant women with poor oral hygiene habits are more susceptible to periodontal complications [6].

Different criteria used for diagnosing periodontitis in pregnant women have been reported [8–15]. Which of these is the best criterion to diagnose periodontitis in pregnant women has not been agreed upon; however, they are important to estimate the frequency of maternal periodontitis and, thus, evaluate the effectiveness of preventive or therapeutic procedures and to determine individual risk [16, 17]. This systematic review evaluated the accuracy of the criteria for diagnosing periodontitis in pregnant women.

## Method

A systematic review of studies of the accuracy of criteria used for diagnosis of periodontitis in pregnant women was carried out, with the protocol registered in the International PROSPERO database—Prospective Register of Systematic Reviews (Registration: CRD42020201471). The wording was based on the PRISMA-P statement—Preferred Reporting Items for Systematic Review and Meta-Analysis Protocols [18].

The question of this research was: How accurate are the different criteria for diagnosing periodontitis in pregnant women? And the acronym used that guided this systematic review was PIROS (P = Population; I = Index test; R = Reference standard; O = Outcome; Study type). The Population (P) was pregnant women. The Index test (I) represented the criteria used to assess the diagnosis of periodontitis in the studies. The Reference standard (R) was the criterion used as the gold standard. Outcome (O) represented the research finding on the impact of a diagnostic strategy. The Study type (S) were validation study designs, diagnostic accuracy. All steps of the study were performed by three independent reviewers and disagreements were resolved among them.

### Search strategies and source of records

According to the acronym PIROS, the search phrase used to identify the condition of interest, periodontitis, was developed as follows: 1) controlled vocabulary terms related to maternal periodontitis were identified (MGP); 2) dental surgeon and periodontist (SSC and ISGF) identified the main keywords and their derivations; 3) retrieved documents were analyzed in detail and those terms that were not related to the diagnosis of periodontitis were discarded, such as the term "gingivitis"; 4) the validated periodontitis search strategy was consulted [19] and 5) the procedure was repeated until the strategy was considered adequate, using PRESS checklist —Peer Review of Electronic Search Strategies [20].

The descriptors used and their respective synonyms were specified by Medical Subject Headings (MeSH). The keywords and Boolean operators in English used in the search strategies were: (*Pregnant Women OR Pregnant Woman OR Woman, Pregnant OR Women, Pregnant OR Pregnancy Complications OR Pregnancy Complications OR Complication, Pregnancy*

*OR Complications, Pregnancy OR Pregnancy Complication) AND (Periodontitis OR Pericementitides OR Pericementitis OR Periodontitides OR Periodontal Diseases OR Disease, Periodontal OR Diseases, Periodontal OR Parodontoses OR Parodontosis OR Periodontal Disease OR Pyorrhea Alveolaris) AND (sensitiv\* OR predictive value\* OR (predictive value\* OR accurac\*)*. Further details of the search strategy are described in the S1 File. The search strategy was similarly adapted for the other databases. To identify periodontitis studies and accuracy, strategies validated by previous studies were used [19, 21].

The research was carried out with no start year of the search, up to 30 March 2024, on search platforms Medline (via PubMed), Embase, Scopus, Web of Science, SciELO, and Lilacs (via Virtual Health Library). Additionally, searches were carried out in the reference lists of the selected articles and the gray literature using the ProQuest and CINAHL databases.

## Selection of eligible studies and data extraction

Among the retrieved studies, only those related to periodontitis in pregnant women were identified. In addition, studies involving animal models were excluded. There was no limitation of language or period of publication. At this stage of the review, the Rayyan application, developed by the Qatar Computing Research Institute, was used [22]. This step was performed by three researchers (SSC, ACMGF, and JETB), after reading titles and abstracts independently. After excluding duplicates, the articles were read in full.

The extraction of data from the included articles was carried out by three researchers (AMH, JMFC, and JSPS) employing the State of the Art through Systematic Review—Start [23] and, subsequently, the information was compared. A standardized electronic spreadsheet was used to record the following extracted data: journal name, authors' names, publication date, study objective, inclusion and exclusion criteria, index test, reference standard, diagnostic values, and main findings.

## Risk of bias

QUADAS-2 instrument: Quality Assessment of Diagnostic Accuracy Studies 2 was used for risk of bias analysis (PML, AOL, and ESS) [24]. This tool comprises 4 domains: patient selection, index test, reference standard, and flow and time. Each domain is assessed for risk of bias. The first three domains are assessed in terms of applicability of the primary studies to our review's research question, in terms of population, tests, and target condition. Several guiding questions help in judging the risk of bias in each domain. The instrument is applied in 4 phases: i) synthesis of the review question, ii) adaptation of the tool and production of specific guidelines for the review; iii) construction of a flowchart of the evaluated studies; and iv) assessment of the risk of bias and applicability.

For the assessment of the risk of bias in each of the four domains, the questions must be answered according to the options "yes", "no", or "unclear". If a study is judged as "low" in all domains related to bias or applicability, it is appropriate to infer an overall judgment of "low risk of bias" or "low applicability concern". If a study is judged to be "high" or "unclear" in one or more domains, it may be judged "at risk of bias" or as having "applicability concerns".

## Data analysis

A descriptive and qualitative analysis of the data was carried out, summarizing the study population, the country of execution, the period of realization, and the frequency of periodontitis for the criteria adopted as test and reference, in addition to the respective diagnostic values. The results were synthesized in a narrative and explanatory chart. It was not feasible to carry out a meta-analysis with the accuracy values of the criteria for diagnosis of periodontitis in

pregnant women, since it was not possible to establish a grouped summary of the diagnostic performance of the tests used in the included studies due to the heterogeneity of periodontitis diagnostic criteria used [25]. Data analysis was performed using the STATA ® version 17 statistical package (Stata Corp LLC, College Station, TX, USA), Serial number: 301706385466.

## Quality of evidence

The quality of the evidence presented was assessed using the GRADE tool—Grading of Recommendations Assessment, Development, and Evaluation, which assigns levels of evidence and classifies the strength of the recommendation for health issues [26].

## Results

592 references were identified by searching the databases. After excluding duplicates, 504 references were screened by reading the title and abstract. A total of 6 articles was chosen for reading in full, only 4 met the eligibility criteria and was included, along with 2 articles found in the manual search, totaling 6 articles included in this study (Fig 1). All these studies were published in the searched databases, with none coming from the gray literature. The description of data extracted from included [27–32] and excluded studies [33, 34], with their respective justifications and references, are summarized in S2 and S3 Files.

### General characteristics of the included studies

A total of 3,005 pregnant women were evaluated in the four investigations, including women aged 18 years or older, with one study including women under the age of 18. The six articles analyzed the validity of different criteria for diagnosing periodontitis, using cross-sectional designs. The studies were conducted in Brazil, Japan, Romania, China, and South Africa (Table 1).

A variety of criteria for diagnosing periodontitis were used. According to the analysis of diagnostic values, three studies used criteria recommended by the Center for Disease Control and Prevention / American Academy of Periodontology, as the gold standard [9, 10], one used

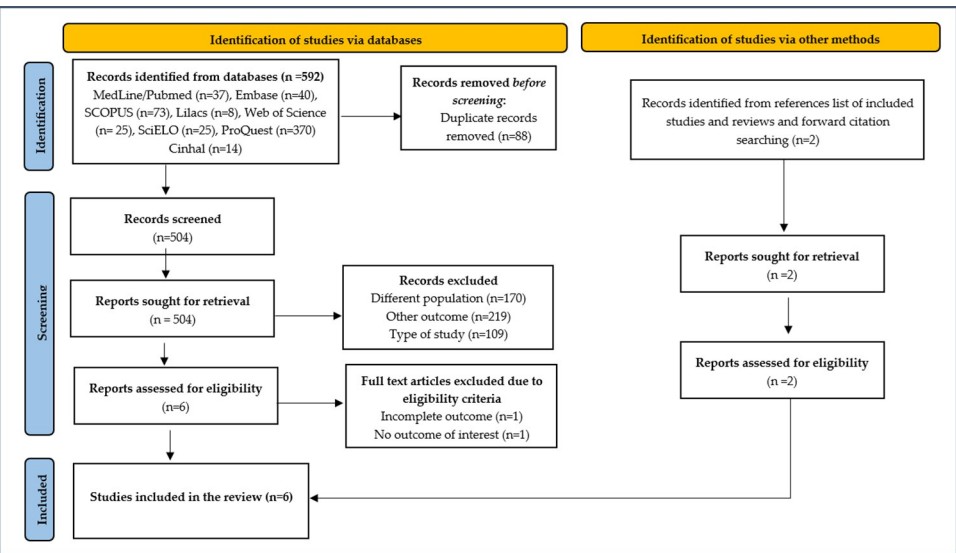

**Fig 1. Flowchart of the study search and selection process.**

**Table 1. General characteristics of the studies used in the systematic review.**

| Characteristics | N | % |
|---|---|---|
| **Study design** | | |
| Cross-sectional | 6 | 100.0% |
| **Sample size** | | |
| ≤ 500 participants | 4 | 66.7% |
| > 500 participants | 2 | 33.3% |
| **Clinical diagnostic criteria for periodontitis used as the gold standard** | | |
| Centers for Disease Control and Prevention/American Academy of Periodontology | 3 | 50.0% |
| Community Periodontal Index of Treatment Needs | 1 | 16.6% |
| Offenbacher et al., (2001) | 1 | 16.6% |
| Gomes-Filho et., (2018) | 1 | 16.6% |
| **Clinical diagnostic criteria for periodontitis used as index tests** | | |
| Self-reported periodontitis | 1 | 11.1% |
| Gomes-Filho et al., (2018) | 1 | 11.1% |
| Albandar et al., (2007) | 1 | 11.1% |
| Bassani et al., (2007) | 1 | 11.1% |
| López et al., (2002) | 1 | 11.1% |
| Nesse et al., (2008) | 1 | 11.1% |
| BANA salivary enzyme test (N-benzoyl-DL-arginine-2-naphthylamide) | 1 | 11.1% |
| American Academy of Periodontics and the European Federation of Periodontics (AAP/EFP): | 1 | 11.1% |
| FDI Periodontal Diseases Chairside Guide (FDI-CG) | 1 | 11.1% |
| **Data collection location** | | |
| Hospital | 2 | 33.3% |
| Healthcare services | 4 | 66.6% |
| **Country of study** | | |
| Brazil | 2 | 33.6% |
| Japan | 1 | 16.6% |
| Romania | 1 | 16.6% |
| South Africa | 1 | 16.6% |
| China | 1 | 16.6% |

the CPITN [15], and the other two used the Offenbacker et al. criterion [8] and Gomes-Filho et al. criterion.

The diagnostic criteria used were self-reported diagnosis, evaluation of the salivary enzymatic test, BANA enzymatic method (N-benzoyl-DL-arginine-2-naphthylamide), and the criteria recommended by Gomes-Filho et al. [13], Albandar et al. [35], Bassani et al. [14], López et al. [12], Nesse et al. [36], AAP/EFP [11] and FDI-CG [37].

Different clinical periodontal parameters used for diagnosing periodontitis were identified in the studies, such as probing depth, clinical attachment level, bleeding on probing, subgingival plaque, and salivary fluid. All investigations evaluated the following diagnostic values: sensitivity, specificity, positive and negative predictive values, and positive and negative likelihood ratios.

## Summary of the results of the included studies

The prevalence of periodontitis varied from 7.7% to 76.7%, based on to the diagnostic criteria used in the included studies. A variety of clinical parameters are used to diagnose periodontitis, making the criteria very heterogeneous among themselves.

**Criteria that employed a combination of probing depth, clinical attachment level, and bleeding on probing.** The study by Conceição et al. (2021), also carried out in Brazil, compared six different criteria for the diagnosis of periodontitis, using periodontal evaluation of all teeth as performed by a trained clinician [27]. The present study included 671 women with a gestational age of 16 to 45 years, with a gestational age of 8 to 32 weeks. Excluding women with twin pregnancies, fewer than four teeth, and diagnosis of any disease requiring antibiotic prophylaxis prior to periodontal examination. Among these criteria, the one proposed by Bassani et al. [14] proved to be a good method for monitoring progression of disease in pregnant women. On the other hand, the criteria by Gomes-Filho et al. [13], López et al. [12], Albandar et al. [35], and Nesse et al. [36] proved to be better for confirming disease, when compared to the reference standard of the Centers for Disease Control and Prevention/American Academy of Periodontology (CDC/AAP) [9, 10]. The criteria by Gomes-Filho et al. [13] and Nesse et al. [36] also showed more greater diagnostic accuracy among the others. These two periodontitis diagnostic methods combined clinical parameters such as probing depth, clinical attachment level, and bleeding on probing.

In 2023, the same researchers conducted a study with the similar objective of validating diagnostic criteria for periodontitis in pregnant women with the same characteristics as the previous study, however, using the criteria of Gomes-Filho et al. [13] as the gold standard [32]. The study included 1,251 pregnant women, who underwent prenatal care in hospitals in two northeastern Brazilian states. Taking the criterion stipulated as the gold standard, which has been used in robust epidemiological studies with populations of pregnant women, it was identified that the criterion recommended by the CDC/AAP [9, 10], Albandar et al. [35] and Bassani et al. [14] tests were considered to have good performance for diagnosing periodontitis in pregnant women in locations that had greater support from the health systems, in order to support screening for the disease. In contrast, the criteria of López et al. [12] and Nesse et al. [36] were best indicated for situations in which there is no possibility of using diagnostic screening methods, but rather confirmation of the disease, that is, in populations with less access to oral health services.

In 2022, a study conducted by Li et al. [31], verified the performance of the classification for diagnosing periodontitis recommended by the AAP/EFP in relation to that stipulated by the CDC/AAP. Furthermore, it explored a practical tool for screening periodontitis in pregnant women. A total of 204 women, in the gestational period between the 7th and 35th gestational week, aged between 24 and 35 years, participated in the study. (i) pregnant women $\geq$ 36 years old were excluded; (ii) edentulous; (iii) using antibiotics for 3 months; (iv) periodontal treatment received within 12 months; (v) self-report of systemically unhealthy illness before pregnancy; and (vi) history of pathological abortion confirmed by medical records. Taking the CDC/AAP criteria as the gold standard, the AAP/EFP classification showed high agreement among pregnant women. The FDI Periodontal Diseases Chairside Guide (FDI-CG), developed in 2018, as a screening tool, was also used as a test for comparison purposes. The authors conclude that the modified FDI-CG scoring system with the addition of gestation stages can serve as a relatively easy-to-use and convenient tool for screening maternal periodontal diseases in general dental practice.

**Self-reported periodontitis.** The research developed by Micu et al. (2020), in Romania, evaluated a self-reported periodontal clinical condition instrument for the diagnosis of periodontitis in 215 pregnant women aged $\geq$ 18 years [29]. The exclusion criteria consisted of i) women aged < 18 years; ii) any systemic disease that could influence the history of periodontitis (e.g. chronic hypertension, pregestational diabetes, chronic inflammatory diseases, etc.); iii) any medical condition requiring antibiotic prophylaxis for dental treatment or systemic antibiotic treatment within the last 3 months; iv) human immunodeficiency virus (HIV) infection.

For that, a 16-item questionnaire was tested containing information about perceived periodontal alterations (in 9 items) and oral hygiene habits (in 7 items). The self-reported periodontitis symptoms section of the instrument contained 9 questions and was created by 3 periodontists native to Romania and, as a model for the questionnaire, the periodontists used a previously validated eight (8) item self-report tool from the CDC/AAP [37]. For this purpose, they used the CDC/AAP criteria, as a reference standard for periodontal diagnosis [9, 10]. A total of 215 pregnant women were evaluated and diagnostic values were analyzed for 4 items of the self-report questionnaire, including gingival swelling, halitosis, previous diagnosis, and treatment of periodontitis. The sensitivity calculated for these four [4] items ranged between 10.8% and 31.1% and the specificity ranged from 83.3% to 97.9. Based on these values, the authors concluded that the self-reported questionnaire failed to accurately detect cases of periodontitis.

**Diagnostic criteria based on dental biofilm.** Turton, Henkel, and Africa evaluated interdental subgingival biofilm between first and second molars or between premolars, testing the hypothesis that the BANA test (N-benzoyl-DL-arginine -2-naphthylamide) could be used by obstetricians and other health professionals to screen for the presence of periodontitis and the risk of adverse pregnancy outcomes in mothers attending prenatal care clinics [30]. A total of 443 pregnant women were evaluated, aged $\geq$ 18 years, in South Africa, using the criteria of Offenbacher et al. as the gold standard [8]. Excluded from the study were smokers; patients with existing heart disease, hypertension, diabetes, asthma or chronic renal disease; mothers with induced labour, previous preterm delivery, or multiple pregnancies; mothers currently using systemic corticosteroids or antibiotics. A total of 282 (64.0%) of the mothers produced a positive BANA test result. The referred index test showed a sensitivity of 86.2%, specificity of 95.1%, positive predictive value of 97.8%, and negative predictive value of 72.6% for predicting periodontitis, proving to be a good test for screening periodontitis in pregnant women.

**Salivary enzymatic diagnostic criteria.** In Japan, Kugahara et al. screened for periodontitis in 221 pregnant women using salivary diagnostic criterion [28]. Women who smoked were excluded from the study. Prior to dental examination, unstimulated whole saliva was evaluated for lactate dehydrogenase (LDH), alkaline phosphatase (ALP), and occult blood. Data were compared with Community Periodontal Treatment Needs Index (CPITN) [15], criterion chosen as the gold standard, obtained after the dental examination. Periodontal examinations are conducted by trained dentists. The diagnostic performance of LDH, ALP, and occult blood was determined in terms of sensitivity, specificity, and the area under the receiver operating characteristics (ROC) curves. Periodontitis was diagnosed in 8.6% of women and gingivitis in 58.4% women. LDH and ALP activity levels were significantly higher in pregnant women with periodontitis when compared to those with gingivitis or healthy periodontium. The occurrence of periodontitis, using the index test and combining salivary LDH, ALP, and occult blood parameters, was 7.7%, with sensitivity of 89.4%, specificity of 62.3%, positive predictive value of 18.2% and negative predictive value of 98.4%. Thus, demonstrating that the aforementioned test proved to be useful for screening pregnant women with periodontitis.

The main findings of the studies included in this systematic review are summarized in Table 2, and further details of these investigations can be found in S2 File.

**Secondary findings.** Other diagnostic evaluations of the tested criteria included the likelihood ratio and predictive values, useful measurements for post-test probability. Although large variability was also observed, most of the criteria tested in this review showed good results, indicating moderate to good quality for post-test probability. Positive predictive values ranged from 18.2% to 98.8%, negative ones from 100% to 25.8%, positive likelihood ratio from 17.68 to 6.03, and negative from 0 to 0.91.

**Table 2. The main findings of the studies included in the systematic review on the accuracy of the criteria for diagnosing periodontitis in pregnant women.**

| Study/Year of Publication | Clinical parameters for diagnosing periodontitis | Gold standard | Index test | Diagnostic accuracy (95% confidence interval) | Authors' conclusions |
|---|---|---|---|---|---|
| Conceição et al., 2023 [32] | Probing depth, clinical attachment level, and bleeding on probing index. | 1. Gomes-Filho et al. [13] | 1 (CRITERION I). Page and Eke (2007), and Eke et al. (2012) [9, 10] United States Centers for Disease Control and Prevention, and American Academy of Periodontology —CDC/AAP 2. Albandar et al. (CRITERION II) [33] 3. Bassani et al. (CRITERION III) [14] 4. Lopez et al. (CRITERION IV) [12] 5. Nesse et al. (CRITERION V) [34] | **1.Sensitivity**: 86.5% % (82.0–90.0) **Specificity**: 72.50% (69.5–75.3) **2. Sensitivity**: 86.5% (82.0–90.2) **Specificity**: 96.0% (73.3–82.4) **3. Sensitivity**: 98.6% (96.5–99.6) **Specificity**: 42.3% (39.2–45.5) **4. Sensitivity**: 70.6% (65.0–81.2) **Specificity**: 96.8% (95.5–97.8) **5. Sensitivity**: 76.5% (71.1–81.2) **Specificity**: 90.6% (88.6–92.4) | CRITERIA I, II and III were considered as good performance tests for diagnosing periodontitis in pregnant women in locations that had greater support from health systems. Criteria IV and V may be better indicated for groups of pregnant women with greater difficulties in accessing oral health services, in general, poorer populations and with a high incidence of periodontitis. |
| Li et al., 2022 [31] | Probing depth, clinical attachment level, and bleeding on probing index. | Page and Eke (2007), and Eke et al. (2012) [9, 10] United States Centers for Disease Control and Prevention, and American Academy of Periodontology—CDC/AAP | 1. American Academy of Periodontics and the European Federation of Periodontics (AAP/EFP) [11]: 2. FDI Periodontal Diseases Chairside Guide (FDI-CG) [37] | **1. Sensitivity**: 100.0% (94.3–100.0) **Specificity**: 92.9% (87.4–96.1) **2. FDI-CG original: Sensitivity**: 20.6% (12.5–32.2) **Specificity**: 99.3% (96.1–100) Version was that pregnancy would be scored with 2 additional points (**FDI-CG all +2): Sensitivity**: 100% (94.3–100) **Specificity**: 61.7% (53.5–69.3) Was that early phase of pregnancy was scored with 1 additional point, and late phase of pregnancy was given 2 additional points (**FDI-CG early phase +1 and late phase +2): Sensitivity**: 92.1% (82.7–96.6) **Specificity**: 70.9% (63.0–78.0) | The AAP/EFP classification is in high agreement with the CDC/AAP definition among pregnant women. The phases of pregnancy integrated FDI scoring system may serve as a convenient screening tool for maternal periodontal diseases in general dental practice. |
| Conceição et al., 2021 [27] | Probing depth, clinical attachment level, and bleeding on probing index. | Page and Eke (2007), and Eke et al. (2012) [9, 10] United States Centers for Disease Control and Prevention, and American Academy of Periodontology—CDC/AAP | 1. Gomes-Filho et al. [13] 2. Albandar et al. [33] 3. Bassani et al. [14] 4. Lopez et al. [12] 5. Nesse et al. [34] | **Sensitivity**: 46.7% (42.0–51.5) **Specificity**: 100% (98.4–100) **Sensitivity**: 62.9% (58.2–67.4) **Specificity**: 96.0% (92.6–98.2) **Sensitivity**: 98.2% (96.5–99.2) **Specificity**: 23.5% (18.1–29.5) **Sensitivity**: 36.0% (31.5–40.6) **Specificity**: 96.5% (93.1–98.5) **Sensitivity**: 37.3% (32.8–42.0) **Specificity**: 99.1% (96.8–99.9) | Bassani et al. criterion had greater sensitivity: ideal for disease screening [14]. The criteria by Gomes-Filho et al [13] and Nesse et al. [34] demonstrated greater specificity, being more appropriate for disease confirmation. |
| Micu et al., 2020 [29] | Probing depth, clinical attachment level. | Page and Eke (2007), and Eke et al. (2012)[9, 10] CDC/AAP. | Self-reported periodontitis: 1. Gum swelling; 2. Halitosis; 3.Previous diagnosis of periodontitis; 4. Pre-treatment. | **Sensitivity**: 31.1% (24.1–38.2) **Specificity**: 83.3% (72.8–93.9) **Sensitivity**: 24.0% (17.5–30.4) **Specificity**: 89.6% (80.9–98.2) **Sensitivity**: 16.8% (11.1–22.4) **Specificity**: 97.9% (93.9–100) **Sensitivity**: 10.8% (6.1–15.5) **Specificity**: 99.9% (93.9–100) | The self-reported questionnaire failed to accurately detect cases of periodontitis. Sensitivity, specificity, positive and negative predictive values were lower when compared to the reference standard. |

(*Continued*)

**Table 2.** (Continued)

| Study/Year of Publication | Clinical parameters for diagnosing periodontitis | Gold standard | Index test | Diagnostic accuracy (95% confidence interval) | Authors' conclusions |
|---|---|---|---|---|---|
| Turton, Henkel, Africa, 2017 [30] | Clinical attachment level, probing depth, and subgingival interdental plaque. | Offenbacher et al. (2001) [8] | BANA (N-benzoyl-DL-arginine-2-naphthylamide) | **Sensitivity:** 86.25% (82.04–89.06) **Specificity:** 95.12% (89.77–97.75) | The test was considered good for screening periodontitis in pregnant women. Sensitivity, specificity, and positive and negative predictive values remain above 80%. |
| Kugahara, Shoseni e Ohashi, 2008 [28] | Probing depth and salivary enzyme. | Community Periodontal Treatment Needs Index (CPITN) [15] | Salivary enzymatic test. | **Sensitivity:** 89.47% (68.91–67.06) **Specificity:** 62.38% (55.52–68.67) | The test was pointed out as useful for screening pregnant women with periodontitis. Sensitivity values and positive and negative predictive values remained above 80%, with the exception of specificity, which obtained a percentage of 62.3%. |

**Assessment of methodological quality.** The QUADAS 2 instrument was used in all reviewed studies and showed good results for the 4 domains evaluated. When assessing the risk of bias, flow and time domains, all studies showed low risk of bias, including for questions related to applicability, such as patient selection, index test, and standard of reference.

For the patient selection domain, there was an uncertain risk of bias in the studies of Kugahara et al. [28] and Micu et al. [29] due to eligibility criteria that were not well defined. The investigations by Conceição et al. [27], Conceição et al. [32], Li et al. [31], and Turton et al. [30] presented an uncertain risk of bias in the index test domain, due to not providing information regarding if the index test results were interpreted without knowledge of the findings of the reference standard test. Micu et al. [29] presented a high risk of bias in the referred domain (index test), because of the use of a self-reported questionnaire and memory bias could be present. The results of the individual evaluations of each study are described in detail in Table 3.

**GRADE system.** To evaluate the degree of certainty of the evidence obtained, the GRADE tool was used, to assess the limitations in the study design, the presence of indirect evidence, inconsistency, and publication bias (Table 4) [26].

**Table 3. Qualitative assessment of cross-sectional studies, evaluating diagnostic criteria for periodontitis in pregnant women, according to QUADAS-2.**

| Study | RISK OF BIAS | | | | CONCERNS ABOUT APPLICABILITY | | |
|---|---|---|---|---|---|---|---|
| | PATIENT SELECTION | INDEX TEST | REFERENCE STANDARD | FLOW AND TIME | PATIENT SELECTION | INDEX TEST | REFERENCE STANDARD |
| Conceição et al., 2023 | * | ? | * | * | * | * | * |
| Li et al., 2022 | * | ? | * | * | * | * | * |
| Conceição et al., 2021 | * | ? | * | * | * | * | * |
| Micu et al., 2020 | ? | ** | * | * | * | * | * |
| Turton, Henkel Africa, 2017 | * | ? | * | * | * | * | * |
| Kugahara, Shosenj, Ohashi, 2008 | ? | * | * | * | * | * | * |

*Low Risk

**High Risk

?Unclear Risk

**Table 4. Factors that reduce the quality of evidence in diagnostic accuracy studies.**

| Factors that determine and may decrease the quality of evidence | Explanations and differences in the quality of evidence for other interventions | Quality of evidence |
|---|---|---|
| **Study design limitations** | All included studies performed diagnostic analysis using cross-sectional designs [1–6]. This item was considered of moderate quality, since in most of the patients included with direct diagnostic uncertainties, the test results were evaluated with appropriate reference standards. It should be noted that a study [4] stipulated as a gold standard a diagnostic criterion for periodontitis without evaluation of the entire mouth, using only index teeth. Additionally, the selection was clearly described, the tests were performed in the same populations of each included study. However, the authors of three studies do not make it clear whether the evaluators of the results were blinded to the evaluation of the index test and the reference standard [1, 2, 5]. | MODERATE |
| **Indirect evidence (Indirectness)** | There is an intrinsic limitation to accuracy studies in this regard, which consists of the lack of direct evidence on the impact of the test on important outcomes for the patient. Authors end up having to make inferences about the balance between the presumed influence of the test on important outcomes and any differences in true and false negatives in relation to complications and testing costs. Thus, low quality is designated for making recommendations due to indirect evidence. | LOW |
| **Inconsistency** | There were inconsistencies regarding diagnostic values (sensitivity, specificity, predictive values and likelihood ratio), especially in two included study [1, 3]. Possibly due to discrepancies in the different criteria for diagnosing periodontitis and the fact that 5 different index tests were analyzed. Similarly, the study that evaluated the self-reported questionnaire also showed differences in analyzing the items: gum swelling, halitosis, previous diagnosis, and treatment of periodontitis. | LOW |
| **Publication Bias** | Although the meta-analysis was not performed, the included studies showed satisfactory sample numbers. Thus, this item was considered as moderate quality. | MODERATE |

**High certainty**: We are very confident that the true effect is close to the effect estimate.

**Moderate certainty**: we are moderately confident in the effect estimate: the true effect is likely to be close to the effect estimate, but there is a possibility that it is substantially different.

**Low certainty**: our confidence in the effect estimate is limited: The actual effect could be materially different from the effect estimate.

**Very low certainty**: We have very little confidence in the effect estimate: the true effect is likely to be substantially different from the effect estimate.

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

Limitations in the study design and publication bias were determined to be of moderate quality of evidence, while items related to indirect evidence and inconsistency were of low quality. The overall evidence rating of the studies were determined to be low, considering the limitations of accuracy of the studies using indirect evidence and absence of direct evidence on patient and discrepancies in diagnostic values (sensitivity, specificity, predictive values, and likelihood ratios).

## Discussion

To the best of our knowledge and according to the latest searches carried out by the authors, this is the first systematic review that compares diagnostic criteria in a group of pregnant women. This review demonstrated that there is a lack of agreement regarding the diagnostic criteria of periodontitis in pregnant women, thus resulting in significant differences in the magnitude of the prevalence of the disease.

## Conflicting validity indicators

In general, there are discrepancies in validity of indicators used in epidemiological studies evaluating periodontitis in pregnant women. Such variances are possibly due to differences in

sample sizes, sampling techniques, disease measurement methods/diagnostic criteria, definitions of periodontitis, socioeconomic status of study population, and follow-up time [38, 39]. Clinical measurements used for diagnosing periodontitis include assessment of probing depth, bleeding on probing, and clinical attachment level [13]. It is known that bleeding on probing is considered an important indicator to suggest the presence of disease in pregnant women, mainly because it is the first clinical sign indicative of inflammation in the periodontal tissue [40]. Gingival inflammation is very common in pregnant women due to hormonal changes that make these women susceptible [41]. However, the combination of all the aforementioned clinical parameters were not routinely used in the identified studies.

The studies by Conceição et al. [27], Li et al. [31], and Micu et al. [29] used the criteria recommended by the US CDC/AAP as the gold standard for the diagnosis of periodontitis [9, 10]. This is considered an internationally recognized criterion, utilizing probing depth and clinical attachment loss as clinical parameters for disease determination, but not including bleeding on probing. With the use of this criterion as a reference standard, high variability was observed in the diagnostic values when compared to the criteria of the index tests.

Recently, several population-based studies have reported the diagnostic performance and agreement between the CDC/AAP system and the AAP/EFP classification [42–44]. However, only the study of Li et al. [31] compared the aforementioned criteria in the population of pregnant women. This study showed that the AAP/EFP classification is in high agreement with the CDC/AAP definition in this population. According to the authors, using as gold standard to the CDC/AAP system, was evaluated by the area under the ROC curve (AUC) for AAP/EFP classification, to distinguish Moderate/Severe periodontitis (Stages II-IV) from No/Mild periodontitis (periodontal health, gingivitis, and Stage I periodontitis), which presented a good value of was 0.979.

According to Conceição et al. [27] and Conceição et al., [32], the combination of all clinical parameters, combined with a complete periodontal examination, may result in more accurate diagnosis of periodontitis. The use of validated and effective criteria for diagnosis, which consider the peculiarities of pregnant women, is not common in clinical practice. Thus, after screening of women potentially affected by periodontitis using more sensitive criteria, the authors recommended the use of more specific criteria to confirm the disease. The adoption of these recommendations in clinical practice may favor earlier identification of periodontitis in pregnant women, and therefore, earlier, and more effective management of the disease.

Self-reported measurements of periodontitis for diagnostic screening do not provide reliable detection of periodontal disease. Micu et al. [29], when testing the diagnostic validity of a self-reported questionnaire for detecting periodontitis, found low accuracy in diagnosing the disease in pregnant women.

The FDI Periodontal Diseases Chairside Guide (FDI-CG), was developed in 2018, as an easy-to-use screening tool in the practice of general dentistry professionals [37]. The scoring systems encompass 7 items for the periodontitis classification profile, namely: age, smoking, diabetes mellitus, tooth loss due to periodontitis, plaque deposits, bleeding on probing, and furcation involvement. Li et al. [31], concluded that in relation to the original version of the scoring system, the modified one with the addition of stages of pregnancy, can serve as a relatively easy-to-use and convenient tool for screening periodontal diseases in pregnant women. However, this instrument was not identified as a widely disseminated criterion for diagnosing periodontitis in epidemiological studies among pregnant women.

Kugahara, Shosenj, Ohashi [28] found good results for screening of periodontitis in pregnant women using a diagnostic method based on salivary enzymes. Salivary biomarkers have been used to detect disease onset and progression, monitor response to therapy and measure

susceptibility to the future progression of periodontitis [45]. Further studies are assessing their applicability in pregnant women and other populations [46].

Turton et al. [30] demonstrated the BANA enzymatic test was a good screening tool for periodontitis screening in pregnant women [47]. However, traditional microbiological evaluations using culture methods, including the isolation and identification of bacterial species, are of little clinical practicality, requiring specific training and of high cost [48].

**Frequency of periodontitis in pregnant women.** High heterogeneity in the prevalence of periodontitis was observed amongst the included studies. Other studies have also revealed wide ranges in the prevalence of periodontitis in pregnant women [49, 50]. The epidemic of periodontal disease during pregnancy has recently been reviewed [17], and a high prevalence of periodontitis in pregnancy has been observed.

It is known that the use of different clinical parameters in the evaluation of the periodontal condition, a lack of periodontal examination of the entire dentition, non-uniformity of diagnostic criteria, differences in study populations including socioeconomic and racial differences, as well as other social determinants of health, can impact on the prevalence and severity of disease [13]. Furthermore, although participants were trained to assess periodontal clinical conditions in two studies, the quality of the examination and frequency of disease could be different if the examinations were carried out by periodontal specialists [27, 28].

New diagnostic criteria, particularly for periodontitis, require validation with evaluation and determination of test accuracy. The use of instruments to assess the quality of published studies is an increasingly encouraged and useful practice for analyzing evidence, especially in the scope of systematic reviews and meta-analyses. When analyzing the QUADAS 2 instrument in the studies included in this systematic review, it was found that none of them presented 100% agreement in relation to the 4 domains evaluated, demonstrating the possibility of bias in the execution of the studies and applicability of the findings. Furthermore, the use of the instrument does not replace careful analysis, from a qualitative point of view, to address concepts and methods used in the studies [51].

**Limitations and strengths.** A major limitation of this review was the small number of primary studies that evaluated clinical diagnosis of periodontitis in pregnant women, which was highly heterogeneous, making it impossible to perform an accuracy meta-analysis and additional analyses.

Despite this, this study was the first systematic review of the accuracy of using different criteria for clinical diagnosis of periodontitis in pregnant women. Robust methodological rigor was used in this review, including validated search strategies for periodontitis, as well as internationally recognized and validated instruments for analysis of risk of bias in accuracy studies [24]. This review found low evidence indicating confidence in the information obtained. This finding is consistent with diagnostic studies since the outcomes measured in these studies are limited only to accuracy outcomes, which behave as substitute outcomes for those that are important for the individual.

In 2018, despite joint efforts by the American Academy of Periodontology and the European Federation of Periodontics to replace the previous classification of periodontal diseases with the Classification of Periodontal and Peri-Implant Diseases and Conditions [11], it is currently not widely used in epidemiological studies, mainly in the group of pregnant women. Important modifications were incorporated, mainly in relation to periodontitis. The "recent" classification requires long-term monitoring and knowledge about the individual's health, making its use in studies of this nature difficult. Therefore, in the present review, only one study was found [31] using the most recent definition of periodontitis, demonstrating that the general profile of periodontal conditions in the pregnant women investigated was similar, using the CDC/AAP [9, 10].

### Implications for practice

The identification of criteria with good diagnostic sensitivity, such as the one by Bassani et al. [14], the BANA [30], and the enzymatic salivary test [28], may favor the screening of periodontitis. In clinical practice, the BANA [30], and the enzymatic salivary test [28] can be good alternatives both for screening and confirming the diagnosis. Tests that present greater specificity and diagnostic accuracy are more appropriate for confirming maternal periodontitis [13, 36]. Therefore, according to the findings of this review, epidemiological studies of an association between two events, e.g., criteria that use a combination of conventional clinical parameters including probing depth, clinical attachment level, and bleeding upon probing, in addition to being more specific, present good levels of reproducibility in research of this nature, such as Gomes-Filho et al. [13].

The use of these criteria for screening and confirming the diagnosis of periodontitis allows for earlier identification of periodontitis in pregnant women, favoring earlier treatment, and, therefore, preventing or reducing potential adverse systemic effects on pregnancy [27, 52].

### Conclusion

The criteria analyzed revealed high variability both in accuracy and prevalence of periodontitis, confirming the lack of consensus of criterion for diagnosing periodontitis in pregnant women. This review clarifies the need for standardization of methods employed for defining periodontitis, with reliable, sensitive, and specific criteria that are sufficient to contribute to the promotion of women's health during the gestational period with an end-goal of reducing maternal-infant mortality and morbidity.

### Supporting information

**S1 Checklist. PRISMA DTA checklist.**
(DOC)

**S1 File. Search strategies with uniterms, and Boolean operators used according to electronic databases.**
(DOCX)

**S2 File. Studies included in the systematic review to evaluate the accuracy of diagnostic criteria for periodontitis in pregnant women.**
(DOCX)

**S3 File. List of excluded studies and reason for exclusion.**
(DOCX)

### Author Contributions

**Conceptualization:** Sarah dos Santos Conceição, Josicélia Estrela Tuy Batista, Simone Seixas da Cruz, Alexandre Marcelo Hintz, Julita Maria Freitas Coelho, Johelle de Santana Passos-Soares, Peter Michael Loomer, Amanda Oliveira Lyrio, Elivan Silva Souza, Ana Cláudia Morais Godoy Figueiredo, Mauricio Gomes Pereira.

**Data curation:** Sarah dos Santos Conceição, Josicélia Estrela Tuy Batista, Simone Seixas da Cruz, Isaac Suzart Gomes-Filho, Alexandre Marcelo Hintz, Julita Maria Freitas Coelho, Johelle de Santana Passos-Soares, Amanda Oliveira Lyrio, Elivan Silva Souza, Ana Cláudia Morais Godoy Figueiredo, Mauricio Gomes Pereira.

**Formal analysis:** Sarah dos Santos Conceição, Josicélia Estrela Tuy Batista, Simone Seixas da Cruz, Isaac Suzart Gomes-Filho, Alexandre Marcelo Hintz, Julita Maria Freitas Coelho, Johelle de Santana Passos-Soares, Peter Michael Loomer, Amanda Oliveira Lyrio, Elivan Silva Souza, Ana Cláudia Morais Godoy Figueiredo, Mauricio Gomes Pereira.

**Investigation:** Sarah dos Santos Conceição, Josicélia Estrela Tuy Batista, Simone Seixas da Cruz, Isaac Suzart Gomes-Filho, Alexandre Marcelo Hintz, Julita Maria Freitas Coelho, Johelle de Santana Passos-Soares, Peter Michael Loomer, Amanda Oliveira Lyrio, Elivan Silva Souza, Ana Cláudia Morais Godoy Figueiredo, Mauricio Gomes Pereira.

**Methodology:** Sarah dos Santos Conceição, Josicélia Estrela Tuy Batista, Simone Seixas da Cruz, Isaac Suzart Gomes-Filho, Alexandre Marcelo Hintz, Julita Maria Freitas Coelho, Johelle de Santana Passos-Soares, Peter Michael Loomer, Amanda Oliveira Lyrio, Elivan Silva Souza, Ana Cláudia Morais Godoy Figueiredo, Mauricio Gomes Pereira.

**Supervision:** Simone Seixas da Cruz, Isaac Suzart Gomes-Filho, Ana Cláudia Morais Godoy Figueiredo.

**Writing – original draft:** Sarah dos Santos Conceição, Josicélia Estrela Tuy Batista, Simone Seixas da Cruz, Isaac Suzart Gomes-Filho, Alexandre Marcelo Hintz, Julita Maria Freitas Coelho, Johelle de Santana Passos-Soares, Peter Michael Loomer, Amanda Oliveira Lyrio, Elivan Silva Souza, Ana Cláudia Morais Godoy Figueiredo, Mauricio Gomes Pereira.

**Writing – review & editing:** Sarah dos Santos Conceição, Josicélia Estrela Tuy Batista, Simone Seixas da Cruz, Isaac Suzart Gomes-Filho, Alexandre Marcelo Hintz, Julita Maria Freitas Coelho, Johelle de Santana Passos-Soares, Peter Michael Loomer, Amanda Oliveira Lyrio, Elivan Silva Souza, Ana Cláudia Morais Godoy Figueiredo, Mauricio Gomes Pereira.

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
