## [Decision Letter · Decision Letter 0]

19 Feb 2024

PONE-D-23-35573Diagnosis of Periodontitis in Pregnant Women: Systematic Review with Prevalence Meta-Analysis of Validity of CriteriaPLOS ONE

Dear Dr. Gomes-Filho,

Thank you for submitting your manuscript to PLOS ONE. After careful consideration, we feel that it has merit but does not fully meet PLOS ONE’s publication criteria as it currently stands. Therefore, we invite you to submit a revised version of the manuscript that addresses the points raised during the review process.

The manuscript is very interesting and has scientific merit. Thank you for the submission to Plos One. However, special attention should be paid to the meta-analysis and the searches carried out, given the possibility of selection bias. I recommend that authors address the issues raised and submit a letter with changes and justifications.

We look forward to receiving your revised manuscript.

Kind regards,

Erika Barbara Abreu Fonseca Thomaz, Ph.D

Academic Editor

PLOS ONE

Journal Requirements:

3. Please include a copy of Table 1,2 and 3 which you refer to in your text on page 8, 13, 14 and 18.

Additional Editor Comments:

Dear authors,

The manuscript has great merit and is within the scope of the Plos One magazine. However, I recommend that you consider the reviewers' observations and others presented below.

INTRODUCTION

Lines 61 to 63: The authors state that “Different criteria used for diagnosing periodontitis in pregnant women have been reported (Offenbacker et al., 2001 (8); Page and Eke (2007) (9) and Eke et al. (2012) (10), Tonetti; Greenwell; Kornman (2018) (11); Lopez et al. (2002) (12); Gomes-Filho et al. (2018) (13); Bassani et al. (2007) (14); Community Periodontal Index of Treatment Needs (15), etc.). Are, in fact, all these diagnostic criteria proposed for use in pregnant women? Do these studies present diagnostic values? Why were these studies not included in the Systematic Review (SR)?

METHOD

Line 78: Insert S= Study design

Line 85 to 109 (Search strategies and source of records): specify that the search strategy was P.O.S., given that the words used in the search referred to population, outcome and study design.

Line 106: was the search updated after April/2023? I recommend updating.

Lines 106 to 109: Why didn't the authors include a Google Scholar search? This database has been widely recommended in Systematic Review studies due to its high sensitivity.

Lines 119 to 122: I recommend including in the qualitative synthesis the age range of the pregnant women included in the primary studies, as well as, when available, the gestational age range of the women when evaluated in these studies. This information is important given the differences in periodontal health depending on age and gestational age. I also recommend mentioning whether the exams included index teeth or the entire mouth, as well as the number of sites on each tooth. As the authors performed a meta-analysis of the prevalence of periodontitis, I suggest that the sample size and sampling design of each primary study be reported.

Line 121: “In addition, studies involving animal models were excluded” – was this the only exclusion criterion? Were studies including high-risk twin pregnancies, women with different comorbidities, smokers, etc. included? Were studies with women at any time during pregnancy included? I suggest specifying the eligibility criteria further.

The systematic review was designed to identify the studies´ accuracy but did not to identify studies that assessed the prevalence of periodontitis in pregnant women. Many prevalence studies were left out of the analyses, so the summary estimate of prevalence may be influenced by selection bias.

RESULTS

The search strategy was likely unable to identify some of the studies that met the inclusion criteria of this work - for example “Li Hui-Jun et al. Diagnostic performance of the AAP/EFP classification and the CDC/AAP case definition among pregnant women and a practical screening tool for maternal periodontal diseases. Journal of Periodontal Research, Volume 57, Issue 5, p. 960-968, 2022”. Suggested other keywords to include in the search:

Self-Assessment

Periodontal Disease

Predictive Value of Tests

Predictive Value of Test

Predictive Values of Tests

Negative Predictive Value

Negative Predictive Values

Predictive Value, Negative

Positive Predictive Value

Positive Predictive Values

Predictive Value, Positive

Diagnostic Errors

Diagnostic Error

Error, Diagnostic

Errors, Diagnostic

Misdiagnosis

Misdiagnoses

Diagnostic Blind Spots

Blind Spot, Diagnostic

Blind Spots, Diagnostic

Diagnostic Blind Spot

Diagnosis

Diagnoses

Diagnose

Diagnoses and Examinations

Examinations and Diagnoses

Diagnoses and Examination

Examination and Diagnoses

I suggest that Figure 2 be included in the text – not as Complementary Material. The estimated prevalences in each study are not presented, as well as the sample size.

No study presented data regarding the area under the ROC curve

DISCUSSION

I recommend discussing the QUADAS-2 instrument used to analyze the risk of bias.

Lines 331 to 342: This section does not discuss the low external validity of these findings, given that the search was not carried out for studies on the prevalence of periodontitis in pregnant women, but for studies of “diagnostic values”.

Reviewers' comments:

Reviewer's Responses to Questions

**Comments to the Author**

1. Is the manuscript technically sound, and do the data support the conclusions?

Reviewer #1: Yes

Reviewer #2: Yes

Reviewer #3: Partly

2. Has the statistical analysis been performed appropriately and rigorously? 

Reviewer #1: Yes

Reviewer #2: I Don't Know

Reviewer #3: Yes

3. Have the authors made all data underlying the findings in their manuscript fully available?

Reviewer #1: Yes

Reviewer #2: Yes

Reviewer #3: Yes

4. Is the manuscript presented in an intelligible fashion and written in standard English?

Reviewer #1: Yes

Reviewer #2: Yes

Reviewer #3: Yes

5. Review Comments to the Author

Reviewer #1: Prezados autores,

Considero artigo excelente, com tema relevante para a pesquisa periodontal, especialmente para os estudos populacionais.

Como recomendação, poderia enfatizar a importância de um sistema de diagnóstico periodontal específico/ mais preciso para a gestante. Para a população em geral, já existe este consenso do método de diagnóstico? Esclarecer na justificativa/ Introdução.

Outo ponto: justificar a presença de um estudo antigo na revisão sistemática (Kugahara, 2008, o qual realizou teste salivar - ainda está em uso este diagnóstico, não seria muito antigo e não usual?).

Reviewer #2: Queries

1. Methods: Kindly include start up and end date of the data search

2. Kindly check the objective/aim in abstract and the goal of the study in the last paragraph in introduction. Both should be same.

3. Also check the research question R1 and R2 and then frame the objectives.

4. Mention about grey literature search and whether the articles from grey literature included or not.

5. Discussion: Add new research results or systematic reviews that validate the use of different diagnostic criteria of periodontitis in pregnant woman.

6. Add new references after adding their results to the discussion.

Reviewer #3: The article aims to present a systematic review of the literature on validation studies of periodontitis diagnosis methods in pregnant women. This is a relevant topic, which can contribute to increasing the quality of periodontal disease screening in this specific population group. In general, the study is well conducted, methodologically adequate and with an appropriate discussion of the results found.

However, in practical terms, the article describes two distinct studies, a systematic review on validation studies and a meta-analysis on the prevalence of periodontal disease in pregnant women, the latter making use of the same database obtained in the systematic review. I think this strategy presents some important biases, which I will comment on below.

The search strategy was clearly aimed at finding validation studies and not cross-sectional studies that would really express the prevalence of periodontal disease in pregnant women. The same can be applied for all other review procedures, such as the methodological quality assessment tool, QUADAS-2, which is specific for evaluate the risk of bias and applicability of primary diagnostic accuracy studies. Therefore, the review will obviously only include studies with such characteristics, which did not necessarily aim to estimate the prevalence of periodontal disease in pregnant women.

Therefore, although the 4 studies that were included are cross-sectional type, they do not present representativeness to illustrate prevalence. A specific systematic review for studies on the prevalence of periodontal disease in pregnant women would certainly find another set of articles.

It seems clear that the authors, upon realizing that it would not be possible to carry out the meta-analysis with the accuracy results, decided to seize the opportunity and use the same database to do so with a focus on prevalence.

However, I think this strategy is not appropriate and can lead to some misguided conclusions. Among them, the one that states that a “high heterogeneity in the prevalence of periodontitis was observed amongst the included studies” (line 332). It is impossible to state that because, as I highlighted previously, the 4 studies included are not representative of the prevalence of periodontal disease in pregnant women. For the same reason, it is not also possible to state that “the summary measurement of periodontitis prevalence was 39.47% [95% Confidence Interval (95%CI): 9.34%-69.6%]” (line 193).

Therefore, I think that the second objective proposed for the study (lines 68 to 69) is not possible to be achieved. I can see two solutions for the article: (1) remove the meta-analysis study, keeping only the systematic review, which, I repeat, is very well done, has relevance and can be published. (2) Redo the meta-analysis based on another literature review, focused on sectional studies on periodontal disease in pregnant women. The second solution seems more complex to me and may make the article excessively long. I think that the results found for the systematic review are already sufficient to justify publication.

6. PLOS authors have the option to publish the peer review history of their article (what does this mean?). If published, this will include your full peer review and any attached files.

Reviewer #1: No

Reviewer #2: No

Reviewer #3: No

---

## [Author Response · Author response to Decision Letter 0]

4 May 2024

Dear Erika Barbara Abreu Fonseca Thomaz, Ph.D

Academic Editor

PLOS ONE

Provided below are the responses to the reviewers’ comments to the manuscript, “Systematic Review to Evaluate Accuracy Studies of the Diagnostic Criteria for Periodontitis in Pregnant Women” (PONE-D-23-35573). 

Journal Requirements:

Please ensure that your manuscript meets PLOS ONE's style requirements, including those for file naming. The PLOS ONE style templates can be found at https://journals.plos.org/plosone/s/file?id=wjVg/PLOSOne_formatting_sample_main_body.pdf and https://journals.plos.org/plosone/s/file?id=ba62/PLOSOne_formatting_sample_title_authors_affiliations.pdf

Response: We would like to thank the reviewer for the instructions.

Response: We would like to thank again the reviewer’s information.

Please include a copy of Table 1, 2, 3, and 4 which you refer to in your text on page 8, 13, 14 and 18.

Response: As suggested, the copy of Tables 1, 2, 3, and 4 were included on pages 7, 8, 13, 14, 15, 16, 17, and 18, 19, respectively.

REVIEWERS’ COMMENTS TO AUTHOR

# Additional Editor Comments:

General Comments

Dear authors,

The manuscript has great merit and is within the scope of the Plos One magazine. However, I recommend that you consider the reviewers' observations and others presented below.

INTRODUCTION

Lines 61 to 63: The authors state that “Different criteria used for diagnosing periodontitis in pregnant women have been reported (Offenbacker et al., 2001 (8); Page and Eke (2007) (9) and Eke et al. (2012) (10), Tonetti; Greenwell; Kornman (2018) (11); Lopez et al. (2002) (12); Gomes-Filho et al. (2018) (13); Bassani et al. (2007) (14); Community Periodontal Index of Treatment Needs (15), etc.). Are, in fact, all these diagnostic criteria proposed for use in pregnant women? Do these studies present diagnostic values? Why were these studies not included in the Systematic Review (SR)?

Response: All of these criteria were used to diagnose periodontitis and were reported in different studies that evaluated the association of periodontitis with undesirable pregnancy outcomes. Therefore, they were not included in the present review as they were not intended to study diagnostic values. To make reading more understandable, the paragraph was rewritten, as follows:

“Different criteria used for diagnosing periodontitis in pregnant women have been reported: Offenbacker et al., 2001 (8); Page and Eke, 2007 (9), and Eke et al., 2012 (10), Tonetti; Greenwell; Kornman, 2018 (11); Lopez et al., 2002 (12); Gomes-Filho et al., 2018 (13); Bassani et al., 2007 (14); Community Periodontal Index of Treatment Needs (15), etc. Which of these is the best criterion to diagnose periodontitis in pregnant women has not been agreed upon; however, they are important to estimate the frequency of maternal periodontitis and, thus, evaluate the effectiveness of preventive or therapeutic procedures and to determine individual risk (16,17).”

METHOD

Line 78: Insert S= Study design

Response: The S= Study design is just at the end of line 78.

Line 85 to 109 (Search strategies and source of records): specify that the search strategy was P.O.S., given that the words used in the search referred to population, outcome and study design.

Response: As recommended, the sentence was included, line 82, as follows:

“Search strategies and source of records

According to the acronym PIROS, the search phrase used to identify the condition of interest, periodontitis, was developed as follows…”

Line 106: was the search updated after April/2023? I recommend updating. 

Response: The search was updated up to 03/30/2024, line 102.

Lines 106 to 109: Why didn't the authors include a Google Scholar search? This database has been widely recommended in Systematic Review studies due to its high sensitivity.

Response: We would like to thank the reviewer for his comment. A total of 8 databases, including gray literature databases, were included in the present systematic review. According to AMSTAR 2, the minimum required would be 4 databases. Therefore, this research was a broad literature search. There are few studies on the topic, a preliminary search on Google Scholar did not bring results that were very different from those found in other databases. However, in future studies, we will take this into account.

Lines 119 to 122: I recommend including in the qualitative synthesis the age range of the pregnant women included in the primary studies, as well as, when available, the gestational age range of the women when evaluated in these studies. This information is important given the differences in periodontal health depending on age and gestational age. I also recommend mentioning whether the exams included index teeth or the entire mouth, as well as the number of sites on each tooth. As the authors performed a meta-analysis of the prevalence of periodontitis, I suggest that the sample size and sampling design of each primary study be reported.

Response: We would like to thank the reviewer’s comment. Information about age and gestational period was added throughout the text.

Line 121: “In addition, studies involving animal models were excluded” – was this the only exclusion criterion? Were studies including high-risk twin pregnancies, women with different comorbidities, smokers, etc. included? Were studies with women at any time during pregnancy included? I suggest specifying the eligibility criteria further.

Response: As mentioned throughout the text of the manuscript, validation studies on the specific topic of periodontitis in pregnant women are not frequent, and this statement is confirmed by the results of the searches carried out. Therefore, in the present systematic review, the only exclusion criteria were studies involving animals. The eligibility criteria for each study were included throughout the text in the description of the results.

The systematic review was designed to identify the studies´ accuracy but did not to identify studies that assessed the prevalence of periodontitis in pregnant women. Many prevalence studies were left out of the analyses, so the summary estimate of prevalence may be influenced by selection bias.

Response: We would like to thank the reviewer’s comment. As the third reviewer suggested, we found it pertinent to remove the prevalence meta-analysis, precisely due to the factors mentioned above.

RESULTS

The search strategy was likely unable to identify some of the studies that met the inclusion criteria of this work - for example “Li Hui-Jun et al. Diagnostic performance of the AAP/EFP classification and the CDC/AAP case definition among pregnant women and a practical screening tool for maternal periodontal diseases. Journal of Periodontal Research, Volume 57, Issue 5, p. 960-968, 2022”. Suggested other keywords to include in the search:

Self-Assessment

Periodontal Disease

Predictive Value of Tests

Predictive Value of Test

Predictive Values of Tests

Negative Predictive Value

Negative Predictive Values

Predictive Value, Negative

Positive Predictive Value

Positive Predictive Values

Predictive Value, Positive

Diagnostic Errors

Diagnostic Error

Error, Diagnostic

Errors, Diagnostic

Misdiagnosis

Misdiagnoses

Diagnostic Blind Spots

Blind Spot, Diagnostic

Blind Spots, Diagnostic

Diagnostic Blind Spot

Diagnosis

Diagnoses

Diagnose

Diagnoses and Examinations

Examinations and Diagnoses

Diagnoses and Examination

Examination and Diagnoses

Response:

We would like to thank the reviewer’s comment. After updating the searches, this study was incorporated into this systematic review, as follows:

“Li Hui-Jun et al. Diagnostic performance of the AAP/EFP classification and the CDC/AAP case definition among pregnant women and a practical screening tool for maternal periodontal diseases. Journal of Periodontal Research, Volume 57, Issue 5, p. 960-968, 2022”.

I suggest that Figure 2 be included in the text – not as Complementary Material. The estimated prevalences in each study are not presented, as well as the sample size.

Response: We would like to thank the reviewer’s comment. However, following the suggestions explained previously, the meta-analysis was removed.

No study presented data regarding the area under the ROC curve

Response: We would like to thank the reviewer’s comment. The studies by Kugahara, Shosenj, Ohashi et al., 2008 and Li et al., 2022 mentioned the ROC curve, which are also mentioned in the manuscript. 

DISCUSSION

I recommend discussing the QUADAS-2 instrument used to analyze the risk of bias.

Response: As recommended, this information was added in the study strengths and limitations, line 444-447. 

Lines 331 to 342: This section does not discuss the low external validity of these findings, given that the search was not carried out for studies on the prevalence of periodontitis in pregnant women, but for studies of “diagnostic values”.

Response: We would like to thank the reviewer’s comment. However, following the suggestions explained previously, the meta-analysis of the prevalence of periodontitis was removed.

#1 Reviewer

General Comments

Dear authors,

I consider it an excellent article, with a relevant topic for periodontal research, especially for population studies.

Comments 

As a recommendation, I could emphasize the importance of a specific/more accurate periodontal diagnosis system for pregnant women. For the general population, does this consensus on the diagnostic method already exist? Clarify in the justification/Introduction.

Response: We would like to thank the reviewer’s comment. There is still no consensus regarding the best diagnostic criteria for periodontitis in pregnant women. This information was added to the justification/introduction, as recommended, as follows:

“Introduction

…

Different criteria used for diagnosing periodontitis in pregnant women have been reported: Offenbacker et al., 2001 (8); Page and Eke, 2007 (9), and Eke et al., 2012 (10), Tonetti; Greenwell; Kornman, 2018 (11); Lopez et al., 2002 (12); Gomes-Filho et al., 2018 (13); Bassani et al., 2007 (14); Community Periodontal Index of Treatment Needs (15), etc. Which of these is the best criterion to diagnose periodontitis in pregnant women has not been agreed upon; however, they are important to estimate the frequency of maternal periodontitis and, thus, evaluate the effectiveness of preventive or therapeutic procedures and to determine individual risk (16,17).”

Another point: justify the presence of an old study in the systematic review (Kugahara, 2008, which performed a salivary test - this diagnosis is still in use, wouldn't it be very old and unusual?).

Response: We believe it is important to include it, even though it is an article from 2008 since salivary markers are considered a good diagnostic method. However, recent articles not carried out in the group of pregnant women prove this statement:

- Zhang Y, Kang N, Xue F, Qiao J, Duan J, Chen F, Cai Y. Evaluation of salivary biomarkers for the diagnosis of periodontitis. BMC Oral Health. 2021 May 17;21(1):266. doi: 10.1186/s12903-021-01600-5. PMID: 34001101; PMCID: PMC8130171.

- Abdullameer MA, Abdulkareem AA. Salivary interleukin-1β as a biomarker to differentiate between periodontal health, gingivitis, and periodontitis. Minerva Dent Oral Sci. 2023 Oct;72(5):221-229. doi: 10.23736/S2724-6329.23.04778-2. Epub 2023 May 10. PMID: 37162330.

-Mohammed HA, Abdulkareem AA, Zardawi FM, Gul SS. Determination of the Accuracy of Salivary Biomarkers for Periodontal Diagnosis. Diagnostics (Basel). 2022 Oct 14;12(10):2485. doi: 10.3390/diagnostics12102485. PMID: 36292174; PMCID: PMC9600931.

#2 Reviewer

General Comments

1. Methods: Kindly include start up and end date of the data search

Response: We would like to thank the reviewer’s comment. The date has been included, line 102-103.

2. Kindly check the objective/aim in abstract and the goal of the study in the last paragraph in introduction. Both should be same.

Response: We would like to thank the reviewer’s comment. Suggestions have been incorporated.

3. Also check the research question R1 and R2 and then frame the objectives.

Response: We would like to thank the reviewer and the research question was checked, as recommended.

4. Mention about grey literature search and whether the articles from grey literature included or not.

Response: In lines 104 and 105, we inform that additionally, searches were carried out in the reference lists of the selected articles and the gray literature using the ProQuest and CINAHL databases.

5. Discussion: Add new research results or systematic reviews that validate the use of different diagnostic criteria of periodontitis in pregnant woman.

Response: We would like to thank the reviewer’s comment. Suggestions have been incorporated.

6. Add new references after adding their results to the discussion.

Response: As recommended, suggestions have been included.

#3 Reviewer

General Comments

The article aims to present a systematic review of the literature on validation studies of periodontitis diagnosis methods in pregnant women. This is a relevant topic, which can contribute to increasing the quality of periodontal disease screening in this specific population group. In general, the study is well conducted, methodologically adequate and with an appropriate discussion of the results found.

However, in practical terms, the article describes two distinct studies, a systematic review on validation studies and a meta-analysis on the prevalence of periodontal disease in pregnant women, the latter making use of the same database obtained in the systematic review. I think this strategy presents some important biases, which I will comment on below.

The search strategy was clearly aimed at finding validation studies and not cross-sectional studies that would really express the prevalence of periodontal disease in pregnant women. The same can be applied for all other review procedures, such as the methodological quality assessment tool, QUADAS-2, which is specific for evaluate the risk of bias and applicability of primary diagnostic accuracy studies. Therefore, the review will obviously only include studies with such characteristics, which did not necessarily aim to estimate the prevalence of periodontal disease in pregnant women. Therefore, although the 4 studies that were included are cross-sectional type, they do not present representativeness to illustrate prevalence. A specific systematic review for studies on the prevalence of periodontal disease in pregnant women would certainly find another set of articles. It seems clear that the authors, upon realizing that it would not be possible to carry out the meta-analysis with the accuracy results, decided to seize the opportunity and use the same database to do so with a focus on prevalence. However, I think this strategy is not appropriate and can lead to some misguided conclusions. Among them, the one that states that a “high heterogeneity in the prevalence of periodontitis was observed amongst the included studies” (line 332). It is impossible to state that because, as I highlighted previously, the 4 studies included are not representative of the prevalence of periodontal disease in pregnant women. For the same reason, it is not also possible to state that “the summary measurement of periodontitis prevalence was 39.47% [95% Confidence Interval (95%CI): 9.34%-69.6%]” (line 193).

Therefore, I think that the second objective proposed for the study (lines 68 to 69) is not possible to be achieved. I can see two

---

## [Editor Report · Decision Letter 1]

9 May 2024

PONE-D-23-35573R1Systematic Review to Evaluate Accuracy Studies of the Diagnostic Criteria for Periodontitis in Pregnant WomenPLOS ONE

Dear Dr. Gomes-Filho,

Thank you for submitting your manuscript to PLOS ONE. After careful consideration, we feel that it has merit but does not fully meet PLOS ONE’s publication criteria as it currently stands. Therefore, we invite you to submit a revised version of the manuscript that addresses the points raised during the review process.

After carefully reading this new version, I consider that the decisions made by the authors made the manuscript more appropriate. I only have a small observation regarding the two new paragraphs inserted at the end of the Introduction, as mentioned below.

We look forward to receiving your revised manuscript.

Kind regards,

Erika Barbara Abreu Fonseca Thomaz, Ph.D

Academic Editor

PLOS ONE

Journal Requirements:

Additional Editor Comments:

Dear authors,

Please, I recommend that you remove the names of the authors of the sentence. Leave only the numerical references. This will make the text cleaner and easier for the reader to understand. The various "commas", "semicolons" and "and" throughout the sentence make the text very confusing. I also suggest that the last paragraph of the Introduction (consisting of just 1 sentence of 2 lines) be joined to the previous paragraph, as follows:

"Different criteria used for diagnosing periodontitis in pregnant women have been reported (8-15). Which of these is the best criterion to diagnose periodontitis in pregnant women has not been agreed upon; however, they are important to estimate the frequency of maternal periodontitis and, thus, evaluate the effectiveness of preventive or therapeutic procedures and to determine individual risk (16,17). This systematic review evaluated the accuracy of the criteria for diagnosing periodontitis in pregnant women."

---

## [Author Response · Author response to Decision Letter 1]

10 May 2024

Dear Erika Barbara Abreu Fonseca Thomaz, Ph.D

Academic Editor

PLOS ONE

Provided below are the responses to the reviewers’ comments to the manuscript, “Systematic Review to Evaluate Accuracy Studies of the Diagnostic Criteria for Periodontitis in Pregnant Women” (PONE-D-23-35573R1). 

Journal Requirements:

Response: The reference list was reviewed, as suggested.

ADDITIONAL EDITOR COMMENTS

Dear authors,

After carefully reading this new version, I consider that the decisions made by the authors made the manuscript more appropriate. I only have a small observation regarding the two new paragraphs inserted at the end of the Introduction, as mentioned below.

Please, I recommend that you remove the names of the authors of the sentence. Leave only the numerical references. This will make the text cleaner and easier for the reader to understand. The various "commas", "semicolons" and "and" throughout the sentence make the text very confusing. I also suggest that the last paragraph of the Introduction (consisting of just 1 sentence of 2 lines) be joined to the previous paragraph, as follows:

"Different criteria used for diagnosing periodontitis in pregnant women have been reported (8-15). Which of these is the best criterion to diagnose periodontitis in pregnant women has not been agreed upon; however, they are important to estimate the frequency of maternal periodontitis and, thus, evaluate the effectiveness of preventive or therapeutic procedures and to determine individual risk (16,17). This systematic review evaluated the accuracy of the criteria for diagnosing periodontitis in pregnant women."

Response: As recommended, the two last paragraphs at the end of the Introduction section were rewritten, as follows:

“INTRODUCTION

…

Different criteria used for diagnosing periodontitis in pregnant women have been reported (8-15). Which of these is the best criterion to diagnose periodontitis in pregnant women has not been agreed upon; however, they are important to estimate the frequency of maternal periodontitis and, thus, evaluate the effectiveness of preventive or therapeutic procedures and to determine individual risk (16,17). This systematic review evaluated the accuracy of the criteria for diagnosing periodontitis in pregnant women.”

---

## [Editor Report · Decision Letter 2]

20 May 2024

Systematic Review to Evaluate Accuracy Studies of the Diagnostic Criteria for Periodontitis in Pregnant Women

PONE-D-23-35573R2

Dear Dr. Gomes-Filho,

We’re pleased to inform you that your manuscript has been judged scientifically suitable for publication and will be formally accepted for publication once it meets all outstanding technical requirements.

Kind regards,

Erika Barbara Abreu Fonseca Thomaz, Ph.D

Academic Editor

PLOS ONE
---

## [Editor Report · Acceptance letter]

9 Jun 2024

PONE-D-23-35573R2 

PLOS ONE

Dear Dr. Gomes-Filho, 

I'm pleased to inform you that your manuscript has been deemed suitable for publication in PLOS ONE. Congratulations! Your manuscript is now being handed over to our production team.

Kind regards, 

on behalf of

Dr. Erika Barbara Abreu Fonseca Thomaz 

Academic Editor

PLOS ONE